# FOXO1 transcription factor modulates airway epithelial responses to viral infection

Nadia M. Daniel[1]*, Ritu Mann-Nüttel[1], Nami Shrestha Palikhe[1], Joaquin López-Orozco[2], Tom Hobman[2], Paul Forsythe[1], Harissios Vliagoftis[1]

1 Division of Pulmonary Medicine and Alberta Respiratory Centre, University of Alberta, Edmonton, Alberta, Canada, 2 Department of Medical Microbiology & Immunology, University of Alberta, Edmonton, Alberta, Canada

* ndaniel@ualberta.ca

## Abstract

The airway epithelium serves as the initial barrier of defence in the respiratory system, guarding against microbial, chemical, and environmental threats introduced through inhaled air. Pattern recognition receptors within the airway epithelium facilitate the detection of these threats. Toll-like receptor-3 (TLR3), a receptor sensitive to double-stranded RNA viruses, plays a vital role in this sensing process. This study focuses on exploring the role of the Forkhead box protein O1 (FOXO1) in airway epithelial cells. While the FOXO1 transcription factor (TF) has been extensively examined in various cell types and diseases, its role in airway epithelial cells is not fully elucidated. FOXO1 expression was altered in the BEAS-2B airway epithelial cell line using a shRNA lentivirus for knockdown and a constitutively active FOXO1 plasmid (CA-FOXO1) for overexpression. Confirmation of FOXO1 knockdown/overexpression was achieved through qRT-PCR, immunofluorescence, and Western blotting. FOXO1 activity was impeded using the FOXO1 inhibitor AS1842856 in BEAS-2B and normal human bronchial epithelial (NHBE) cells. TLR3 expression was assessed through qRT-PCR and Western blot. Inflammatory cytokines/chemokines IL6, CXCL10, TSLP, CCL26, IL8, GM-CSF, IFN-λ1, TNF-α and CCL2 were analyzed using MSD Immunoassays after stimulation with TLR3 ligand Poly(I:C). ECIS analysis demonstrated that FOXO1-deficient airway epithelial cells exhibit enhanced recovery of barrier integrity following wounding, with faster restoration and higher resistance compared to control cells. FOXO1-deficient BEAS-2B cells exhibited reduced TLR3 mRNA expression while cells transfected with constitutively active FOXO1 displayed increased TLR3 mRNA expression, without corresponding changes in TLR3 protein levels. Inhibition of FOXO1 activity reduced TLR3 mRNA expression in BEAS-2B and NHBE cells. Co-treatment of BEAS-2B cells with the FOXO1 inhibitor and Poly(I:C), resulted in lower IL6 and CCL2 release compared to stimulation with Poly(I:C) alone, but did not affect the release of the other cytokines/chemokines measured. Finally, Poly(I:C) stimulation induced a time-dependent increase in FOXO1 nuclear localization in

**Data availability statement:** All relevant data are within the manuscript and its Supporting information files.

**Funding:** This work was supported by a Discovery grant from NSERC and the GSK/CIHR Chair in Airway Inflammation to HV. NMD was supported by an NSERC Canada Graduate Scholarship Doctoral Program studentship. PF is the AstraZeneca (Canada) Inc., Chair in Asthma and Obstructive Lung Disease.

**Competing interests:** The authors have declared that no competing interests exist.

airway epithelial cells. FOXO1 depletion had no effect on RIG-I, MAVS, or MYD88 expression, suggesting selective regulation of TLR3 among antiviral RNA-sensing pathways. FOXO1 inhibition in SARS-CoV-2–infected NHBE cells significantly reduced viral spike RNA levels 24 h post-infection. Furthermore, we showed that FOXO1 knockdown did not affect cell proliferation, or cell death. In-silico analysis suggested that FOXO1 can bind to the TLR3 promoter, but our EMSA data were inconclusive. These findings indicate that FOXO1 selectively modulates airway epithelial inflammatory and barrier responses. FOXO1 inhibition may have therapeutic potential in mitigating airway inflammation. However, further studies are needed to elucidate the underlying mechanisms of FOXO1-mediated TLR3 regulation.

## Introduction

The airway epithelium is the first defence mechanism against microbial, chemical, and other environmental insults that enter the body through inhaled air. The epithelium protects by creating a semi-permeable physiological barrier between the organism and the external environment [1,2]. Forkhead box O (FOXO) transcription factors are broadly involved in regulating cellular homeostasis, stress responses, and immune function across various tissues. Amid the intricate host-virus interactions of the respiratory system, Forkhead box O1 (FOXO1) contributes to epithelial homeostasis and modulates immune responses to viral infections in the airways [3,4]. Understanding these interactions is crucial, as viral infections may significantly exacerbate the severity and progression of chronic respiratory conditions.

Airway epithelial cells (AECs) do not merely form a physical barrier, but are active participants in immune surveillance and regulation [5]. Beyond their structural role, these cells express a repertoire of receptors, including Toll-like receptors (TLRs), such as TLR3 and TLR4, enabling them to sense and respond to microbial challenges. Specifically, TLR3 serves as a sentinel of the innate immune system, specializing in detecting double-stranded RNA generated during the replication of viruses [6]. TLR3 on AECs acts as a first line of defence against respiratory viral infections. Upon activation, TLR3 triggers a cascade of signalling events that culminate in the activation of immune responses, contributing significantly to the host's defence mechanisms [7–9].

AECs express the FOXO1 transcription factor, which regulates key processes such as oxidative stress responses, epithelial repair, and immune modulation. Despite significant advancements in understanding FOXO1 function, persistent knowledge gaps remain, revealing the complexity of its signalling pathways [10–12]. Palumbo et al., 2017 demonstrated FOXO1's involvement in oxidative stress regulation in lung epithelial cells [10]. Seiler et al., 2013 showed that FOXO1 contributes to epithelial regeneration post-injury [11]. More recently, a preprint by Uliczka et al., 2024 identified FOXO1 as a key modulator of airway inflammation, influencing cytokine production [12]. Despite these studies, the functional implications for chronic airway diseases remain unresolved. These

findings suggest that FOXO1 operates within a broader, yet poorly characterized, regulatory network that governs airway homeostasis. Further research is needed to clarify its interactions with immune signalling pathways, such as TLR3-mediated responses.

Previous data in our lab has shown that FOXO1 is expressed in bronchial epithelial cells and involved in the regulation of Protease-Activated Receptor 2 (PAR2), an important receptor upregulated in asthmatics [13]. Building on our prior research demonstrating FOXO1's regulation of PAR2, we sought to unravel additional facets of FOXO1's functionality. A study by Kim et al., 2019 highlighted FOXO1's involvement in orchestrating the migratory response of human Mesenchymal Stromal Cells (hMSCs) following TLR3 stimulation [14]. Given AECs' known migratory capabilities during wound healing, we investigated whether FOXO1 is also involved in barrier function.

## Materials and methods

### Cell culture

NHBE cells derived from five donors were purchased from Lonza (Walkersville, MD), BEAS-2B cells were purchased from ATCC (Manassas, VA) and were cultured as previously described [13]. BEAS-2B and NHBE cells were treated with FOXO-1 inhibitor AS1842856 (1 µmol/L; Calbiochem, San Diego, CA) or Poly(I:C) (50 µg/mL; Millipore Sigma) for 24 hours.

### FOXO1 deficient BEAS-2B cells

Stable FOXO1 deficient BEAS-2B cells (FOXO1 shRNA) were generated by transducing cells with lentivirus (MOI of 10) encoding an shRNA (GATAACTCGACTTATTGTCCTGTTTTTGCCGGGCCGGAG TTTAGCCAGTCCAACTCGA) against FOXO1 mRNA in MISSION® pLKO—1-puro plasmid (Sigma-Aldrich). As a negative control, a lentivirus expressing scrambled shRNA, which does not target any mammalian gene, was used to transduce BEAS-2B cells (scrambled shRNA) (Addgene #1864). FOXO1 deficient and scrambled shRNA cells were selected by adding 0.5 µg/mL of puromycin to the media. Cell lines were considered stable after three passages in the presence of puromycin and were then used for five consecutive passages for our experiments.

### CA-FOXO1 BEAS-2B cells

BEAS-2B cells were transfected with a plasmid containing a constitutively active FOXO1 mutant (CA-FOXO1) (HA-Foxo1ADA (pCMV5) #12143, Addgene) using Mirus Bio TransIT-LT1 (Thermo Fisher Scientific, Waltham, MA). This mutant contains alanine substitutions at the three canonical Akt phosphorylation sites (Thr24, Ser256, Ser319), rendering FOXO1 resistant to Akt-mediated phosphorylation, nuclear export, and degradation. As a result, CA-FOXO1 remains transcriptionally active in the nucleus independent of upstream PI3K–Akt signaling [15]. The CA-FOXO1 plasmid was sequenced, and the identification was confirmed with Plasmidsaurus.

### RNA extraction, reverse transcription, and quantitative RT-PCR

RNA extraction was performed using TRIzol per manufacturer's protocol (Thermo Fisher Scientific, Waltham, Mass); 0.5 µg of RNA was reverse transcribed with 5µM Oligo (dt) 12−18 (Life Technologies) and 200U M-MLV reverse transcriptase (Life Technologies) in a 20 µL final volume. Gene expression was analyzed using TaqMan assays (Thermo Fisher Scientific, Waltham, MA) for FOXO1 (Hs00231106_m1), FOXO3 (Hs00818121_m1), FOXO4 (Cat#Hs00172973_m1), TLR3 (Hs01551078_m1), RIG-1/DDX58 (Hs01061436_m1), MAVS (Hs00920075_m1), MYD88 (Hs01573837_g1) and GapdH (Hs02758991_g1) as internal control. In each experiment, FOXO1-deficient cell lines were compared against two controls: scrambled shRNA as the experimental control and untreated BEAS-2B cells. All data was normalized to the untreated BEAS-2B cell line (control).

NHBE cells were infected for 24 hours with SARS-CoV-2 (MOI = 0.5 – Strain Canada/ON/VIDO-01/2020 – GISAID: EPI ISL_425177). Total RNA was isolated from NHBE cells using the RNA NucleoSpin Kit (Machery Nagel) following manufacturer's instructions. cDNA was synthesized using random primers (Invitrogen) and Improm-II reverse transcriptase (Promega). qPCR was performed using primers from Integrated DNA Technologies and PerfecTa SYBR Green SuperMix with Low ROX (Quanta Biosciences) on a Bio-Rad CFX96 system. The gene targets and primer sequence are listed in the table below. The CT values were normalized using *Actb* mRNA as the internal control.

Primers used for qRT-PCR.

Target Gene Primer sequences (5′→3′)

SARS-CoV-2 spike Fwd: CCTACTAAATTAAATGATCTCTGCTTTACT
Rev: CAAGCTATAACGCAGCCTGTA
ACTB Fwd: CCTGGCACCCAGCACAAT
Rev: GCCGATCCACACGGAGTACT

## Confocal microscopy

BEAS-2B cells were cultured on collagen-coated glass coverslips, treated with or without 50 µg/mL Poly(I:C), a TLR3 ligand, for 24 hours. Cells were fixed with 4% paraformaldehyde diluted in PBS and permeabilized with 0.5% Triton X100–PBS. After blocking with 1% BSA-PBS, cells were incubated with anti–human FOXO1 rabbit mAb (C29H4 #2880, Cell Signaling Technology) overnight at 4°C, followed by incubation with Donkey anti-rabbit IgG AF555 secondary antibody (#A-31572, Invitrogen) for 30 minutes at room temperature. Cells were incubated with DAPI to stain nuclei (Sigma-Aldrich) and Phalloidin AF488 (#A12379, Invitrogen) to stain the cytoskeleton. Coverslips were mounted in ProLong Gold (P36930, Molecular Probes, Eugene, Ore). Cells stained with isotype or only secondary antibody were used as negative controls. All images were obtained on an Olympus IX81 epifluorescence microscope and analyzed using CellSens software (Olympus Canada, Mississauga, ON, Canada) and Volocity image analysis software (version 6.3, Puslinch, ON, Canada) as previously described [16]. The mean FOXO1 fluorescence intensity was quantified for both nuclear and whole-cell compartments, with DAPI colocalization defining the nucleus and phalloidin staining outlining cell boundaries respectively. Statistical Analysis with Shapiro-Wilk normality test + Kruskal-Wallis multi-comparison test was used with a Dunn correction for nonparametric datasets * $p < 0.05$.

## Western blotting

Cell lysates were separated on a 12% SDS-PAGE gel and proteins were then transferred to a nitrocellulose membrane. Membranes were incubated with anti–FOXO1 mAb (Cell Signaling Technology, #2880) or anti-TLR3 mAb (Cell Signaling Technology, #6961) primary antibodies, followed by IRdye-conjugated goat anti-rabbit IgG (LI-COR, Lincoln, Neb); membranes were imaged with the Odyssey Infrared Imager (LI-COR). Densitometry analysis was used in Image Studio Lite software (LI-COR) to analyze the images, with the results presented as the ratio of the target gene to β-actin. Primary mouse anti-β-actin mAb (Santa Cruz Biotechnology, SC-69679) and IRdye-conjugated donkey anti-mouse IgG (LI-COR, Lincoln, Neb) were used as a loading control.

## Cell proliferation assay

The WST-1 assay was used per the manufacturer's standard protocol to measure cell proliferation through metabolic activity. Briefly, BEAS-2B cells were cultured overnight on a 96-well plate. Cells were incubated with WST-1 for 2h at 37°C in a humidified $CO_2$ incubator. The absorbance was measured at 450 nm. Cells were also counted using a hemocytometer following Trypan Blue (Thermo Fisher Scientific, Waltham, MA, #15250061) exclusion to assess viability and total cell numbers.

## Apoptosis assay

Apoptosis and cell death were studied by staining FOXO1 deficient cells for Annexin V and PI dye (Thermo Fisher Scientific, Waltham, MA, #88–8007) using Flow cytometry. Dead cells were identified as BEAS-2B that were double positive: Annexin V+/PI+). Apoptotic cells were identified as BEAS-2B cells positive for Annexin V only.

## Electric cell-substrate impedance sensing (ECIS)

Barrier function and wound healing capacity of BEAS-2B cell monolayers were assessed using an electric cell-substrate impedance sensing system (ECIS 1600, Applied BioPhysics, Troy, NY) as described previously [17]. Briefly, BEAS-2B cells were seeded onto collagen-coated ECIS 8-well arrays (8W1E PET) and placed in a humidified $CO_2$ incubator. Resistance at 4000 Hz was used as the primary readout of monolayer integrity, based on established sensitivity to cell-cell junctions and adhesion properties. After establishing stable baseline resistance values, a controlled electrical wound was applied using a 5.0 V signal at 4000 Hz for 60 seconds to disrupt the monolayer. Resistance measurements were then recorded every 5 minutes to monitor the restoration of barrier integrity during the recovery phase. All resistance values were normalized to each well's pre-wounding baseline to ensure comparability across conditions and account for variation in initial monolayer characteristics.

## Cytokine measurements

Cells were treated with or without 50 µg/mL Poly(I:C) for 24 hours, and supernatants were collected. Samples were analyzed in triplicate using Meso Scale Discovery Multiplex Assay kits (MSD, Rockville, MD, USA) following the manufacturer's protocols. Customized V-Plex human Cytokine Panel 1 (K15050D-1) multiplex kit was used for CXCL10. U-plex Biomarker I (K15067M-1) multiplex kits were used for Thymic Stromal Lymphopoietin (TSLP), CCL2, CCL26, Interleukin-8 (IL-8), Interleukin-6 (IL-6), Granulocyte-macrophage colony-stimulating factor (GM-CSF), interferon λ1 (IFN-λ1) and Tumour Necrosis Factor α (TNF-α).

## In silico analysis

ChIP-Seq data were retrieved from Gene Expression Omnibus and analyzed. ChIP-Seq data were retrieved using the ChIP-Atlas (PMID: 35325188) and analyzed in the Integrative Genome Browser (https://doi.org/10.1093/bib/bbs017). The FOXO1 DNA binding motif was taken from the HOCOMOCO database (https://doi.org/10.1093/nar/gkx1106).

## Electrophoretic mobility–Shift assay (EMSA)

Nuclear extracts were prepared from BEAS-2B cells per protocol using a NE-PER Nuclear and Cytoplasmic Extraction Kit (Thermo Fisher Scientific, Waltham, MA, # 78833). Protein concentration was determined using the BCA assay, and ~30–50 µg nuclear protein was used per EMSA reaction. Binding reactions (20 µl) contained 5 µl of 4×EMSA binding buffer (40 mM HEPES pH 7.9, 20% glycerol, 100 mM NaCl, 8 mM DTT, 0.4 mM EDTA, 0.2 µg/µl poly[dI-dC]), nuclear extract, and water to volume. Following a 10-min pre-incubation at room temperature, 1 pmol IRDye700-labeled double-stranded oligonucleotide containing the FOXO1-binding motif 'TGTTT' (forward 5' TAAAAACTAGGTGTTTTTCAGAGGCGGTTT 3'; and reverse 5' AAACCGCCTCTGAAAAACACCTAGTTTTTA 3') was added, and samples were incubated for an additional 20 min. Specificity was assessed using a>50-fold molar excess of unlabeled probe (cold-competitor). Complexes were resolved on 6% Tris-glycine-EDTA (TGE) polyacrylamide gels pre-run at 105 V for 15 min. Electrophoresis proceeded at 105 V for 35–45 min until the dye front migrated approximately three-quarters down the gel. Gels were imaged using a Li-Cor Odyssey scanner.

## Statistics

Results are expressed as means ± SEMs. Unless stated otherwise in the Fig legends, the following statistics were used. For comparisons between 2 groups, the 2-tailed paired t-test was performed for AEC experiments. For 3 or more groups,

an ANOVA with the Tukey multiple comparison test was used. Differences in nonparametric data (i.e., cell size) were analyzed by Kruskal–Wallis with a post hoc analysis by Dunn's tests. Statistical analysis was performed using GraphPad Prism version 10 on Windows (GraphPad Software, La Jolla, California, USA). A p-value <0.05 was considered statistically significant.

## Results

### Reduced FOXO1 expression alters barrier recovery after epithelial injury in BEAS-2B cells

FOXO1 deficient BEAS-2B cells were created using a shRNA lentivirus, and FOXO1 knockdown was confirmed at the mRNA and protein levels. BEAS-2B cells transduced with the FOXO1 shRNA lentivirus showed >90% reduction of FOXO1 mRNA expression by RT-qPCR **(Fig 1A)** and >65% decrease in FOXO1 protein expression by Western blot (Fig 1B and 1C) compared to cells transduced with a scrambled shRNA lentivirus. Downregulation of FOXO1 did not affect the expression of FOXO3 or FOXO4 mRNA in BEAS-2B cells **(Fig 1D+E)**, indicating no compensatory changes in other FOXO isoforms. Immunofluorescence imaging showed reduced FOXO-1 protein presence in the nucleus of FOXO1-deficient BEAS-2B compared to control cells **(Fig 1F+G)**, suggesting that shRNA transduced cells have lower levels of active FOXO1.

We then studied the effect of FOXO1 downregulation on BEAS-2B cell proliferation, apoptosis and barrier integrity, key physiological processes in which FOXO1 has been implicated across different cell types. FOXO1 downregulation had no effect on BEAS-2B proliferation as measured by a metabolic assay (Fig 2A) or by total cell counts at the end of incubation **(Fig 2B).** No difference was found between FOXO1 deficient BEAS-2B and scrambled shRNA for either apoptosis or cell death after 5 days of culture **(Fig 2C+D+E)**.

Regarding barrier function, both FOXO1-deficient and scrambled shRNA BEAS-2B showed an initial sharp decline in resistance following wounding **(Fig 2F)**. FOXO1-deficient cells consistently showed higher resistance throughout the recovery phase compared to scrambled shRNAs **(Fig 2F),** indicating faster and more effective restoration of barrier integrity.

To quantify the recovery, we measured the change in resistance post-wounding. FOXO1 knockdown cells exhibited a greater increase in resistance than control cells **(Fig 2G)**, supporting the observation of enhanced repair capacity. Furthermore, endpoint resistance measurements at the conclusion of the experiment were also significantly higher in FOXO1-deficient cells **(Fig 2H)**. These findings suggest that FOXO1 may function as a negative regulator of epithelial repair, and its downregulation facilitates a more efficient recovery of barrier function following injury.

### FOXO1 modulates TLR3 mRNA expression in BEAS-2B cells

FOXO1 has been implicated in regulating immune and stress-responsive transcriptional programs in multiple cell types [4]. FOXO has been implicated in regulating TLR-dependent responses in other cell types, including responses to TLR3 (viral) [14] and TLR4 expression (bacterial) [18]. Therefore, we examined whether FOXO1 influences expression of pattern recognition receptors involved in epithelial responses during infection. RT-qPCR analysis showed a significant reduction in TLR3 mRNA levels in FOXO1 deficient cells compared to scrambled shRNA, while TLR4 expression remained unchanged (Fig 3A+B). Western blot analysis demonstrated that FOXO1 knockdown did not significantly alter TLR3 protein expression compared to scrambled control cells (Fig 3C+3D).

To test whether persistent FOXO1 activity is sufficient to influence TLR3 transcription, we generated a FOXO1 overexpression model using a constitutively active FOXO1 (CA-FOXO1) mutant. Confocal immunofluorescence confirmed higher FOXO1 nuclear localization 24 h after CA-FOXO1 transfection compared to vector control cells (Fig 4A+B), validating increased FOXO1 activity. CA-FOXO1-transfected BEAS-2B cells showed a significant increase in TLR3 mRNA expression compared to control vector transfected cells (Fig 4C). However, western blot analysis showed no change in TLR3 protein expression (Fig 4D+E).

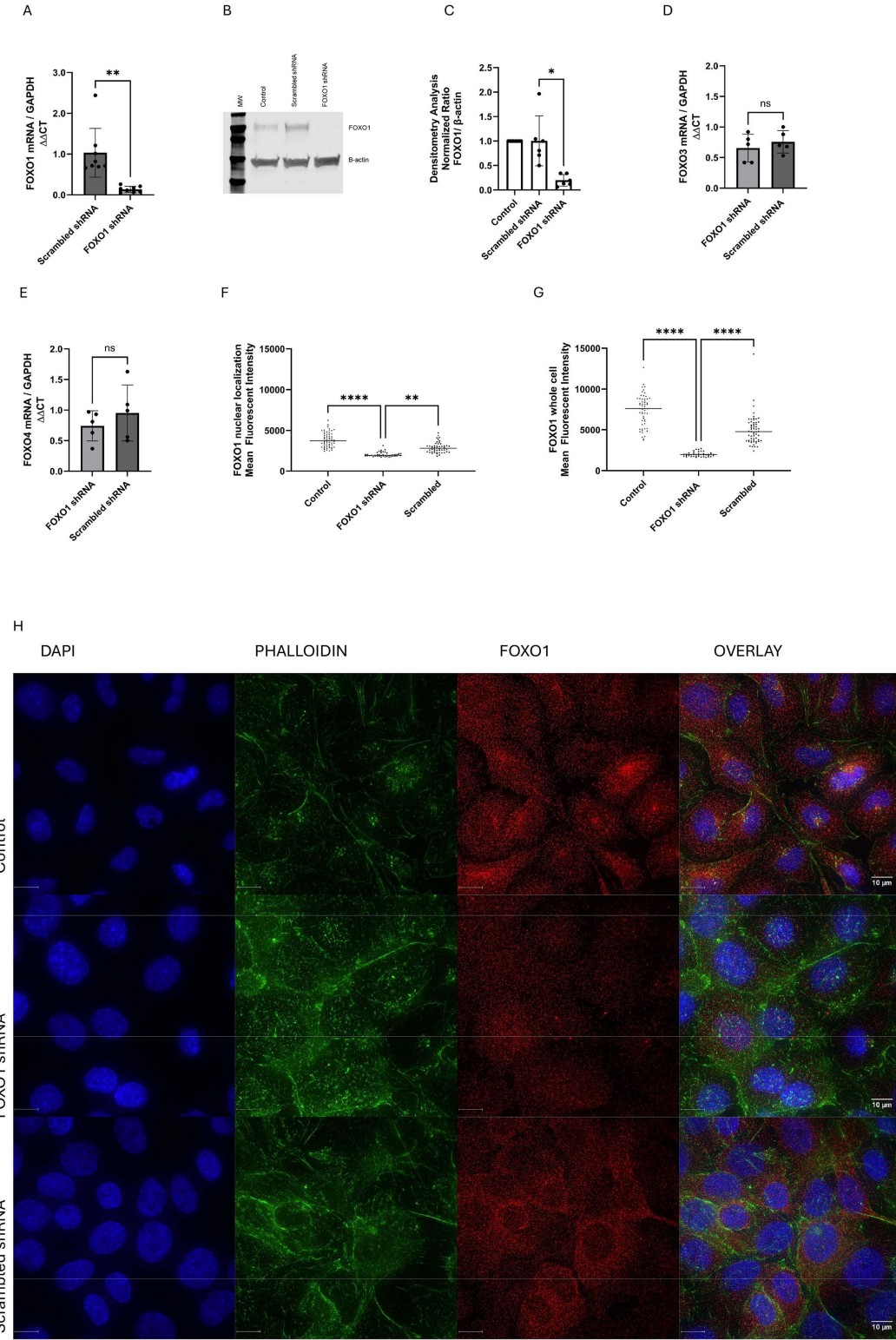

**Fig 1. FOXO1 expression in BEAS-2B cells transduced with FOXO1 shRNA lentivirus or scrambled shRNA lentivirus.** A) FOXO1 mRNA expression was analyzed by RT-qPCR in BEAS-2B cells transduced with FOXO1 shRNA or scrambled shRNA (n = 8). B) Representative Western blot and C) densitometry quantification (n = 6) of FOXO1 protein levels in cells treated with FOXO1 shRNA lentivirus, scrambled shRNA lentivirus and untransduced cells (control). Statistical analysis with t-test *p < 0.05, **p < 0.01. FOXO3 (D) and FOXO4 (E) expression levels in FOXO1 shRNA and scrambled shRNA

lentivirus transfected cells were assessed by RT-qPCR (n = 5). Mean fluorescence intensity of FOXO1 localized to the nucleus (F) and whole cell expression (G) was quantified with Volocity software. For each group 40-60 cells per slide were analyzed. Statistical Analysis was performed using Anova **P < 0.01, **** P < 0.0001 H) Immunofluorescence staining for FOXO1 in BEAS-2B cells transduced with FOXO1 shRNA or scrambled shRNA. FOXO1 (red) was detected using an anti-FOXO1 monoclonal antibody, nuclei were stained with DAPI (blue), and the cytoskeleton was visualized with phalloidin (green). Images were captured at 60 × magnification using a DeltaVision Confocal Microscope.

To further support a role for FOXO1 in TLR3 transcriptional regulation, we treated BEAS-2B cells with the FOXO1 inhibitor AS1842856, which selectively binds the active form of FOXO1 and decreases its transcriptional activity [19]. To determine the IC50 for FOXO1 inhibition, a dose-response experiment was conducted using concentrations of 0.01 μM, 0.1 μM, and 1.0 μM of AS1842856 (S1G Fig); 1.0 μM AS1842856 produced consistent inhibition and was used for subsequent experiments. In unstimulated BEAS-2B cells, 1.0 μM AS1842856 treatment led to a significant reduction in TLR3 mRNA expression (Fig 3M).

## Lack of FOXO1 activity decreases IL6 and CCL2 release in response to TLR3 activation

We stimulated BEAS-2B cells with 50 μg/mL Poly(I:C) and showed an increase in TLR3 mRNA expression, consistent with previous reports [20]. The addition of 1.0 μM FOXO1 inhibitor resulted in a decrease in Poly(I:C)- induced TLR3 mRNA expression in BEAS-2B (Fig 3M). To test whether the same is true for primary human bronchial epithelial cells, we activated NHBE cells with Poly(I:C) in the presence or absence of 1.0 μM of the FOXO1 inhibitor. Consistent with BEAS-2B results, Poly(I:C) stimulation induced TLR3 mRNA expression in NHBE cells, and this induction was markedly reduced in the presence of the FOXO1 inhibitor (Fig 3N).

The above data suggest that FOXO1 activation mediates TLR3 effects in airway epithelial cells. To validate our hypothesis that FOXO1 is activated during epithelial responses to TLR3 activation, we examined FOXO1 nuclear localization following Poly(I:C) stimulation in BEAS-2B airway epithelial cells. Under unstimulated conditions, FOXO1 nuclear fluorescence was detected at low levels in resting cells, and there was a time-dependent increase in FOXO1 nuclear localization after Poly(I:C) stimulation (Fig 5A) that peaked at 3 h and was back to baseline by 6 h (Fig 5B). These results strongly suggest that Poly(I:C) activates FOXO1 in airway epithelial cells.

To assess whether FOXO1 influences other functional responses downstream of TLR3 activation, cells were stimulated with Poly(I:C), and cytokine release was measured. Poly(I:C)-mediated BEAS-2B activation increased the release of 7 out of 8 cytokines analyzed (TSLP, CCL2, CXCL10, CCL26, IL8, IL6, IFN-λ and TNF-α) but did not induce GM-CSF release following 24 h stimulation. FOXO1-deficient cells showed selective impairment in IL6 and CCL2 release by TLR3 activation (Fig 3E + F), while the release of the other cytokines remained unaffected (Fig 3G–3L). In contrast, FOXO1 downregulation did not affect baseline release of cytokines from BEAS-2B cells (Fig 3E–3L). This data supports a role for FOXO1 in regulating TLR3 transcriptional responses in airway epithelial cells.

To determine whether FOXO1 affects other viral RNA sensing pathways, we measured expression of mRNA for DDX58 (encoding RIG-I), its adaptor MAVS, and MYD88 (an adaptor for TLRs other than TLR3) in BEAS-2B cells after FOXO1 knockdown. RT-PCR analysis at baseline and after Poly(I:C) stimulation (8 h and 24 h) showed no significant differences in RIG-I, MAVS, or MYD88 transcript levels between FOXO1 knockdown and control cells (Fig 4F + G + H), indicating that FOXO1 does not alter transcription of these components under our experimental conditions.

To assess the relevance of FOXO1 in epithelial responses during an active viral infection, primary NHBE cells were infected with SARS-CoV-2 in the presence or absence of the FOXO1 inhibitor *AS1842856*. Total RNA collected at 24 h post-infection showed a significant reduction in viral spike RNA levels in FOXO1-inhibited cells compared with DMSO-treated cells (n = 3) (Fig 4I).

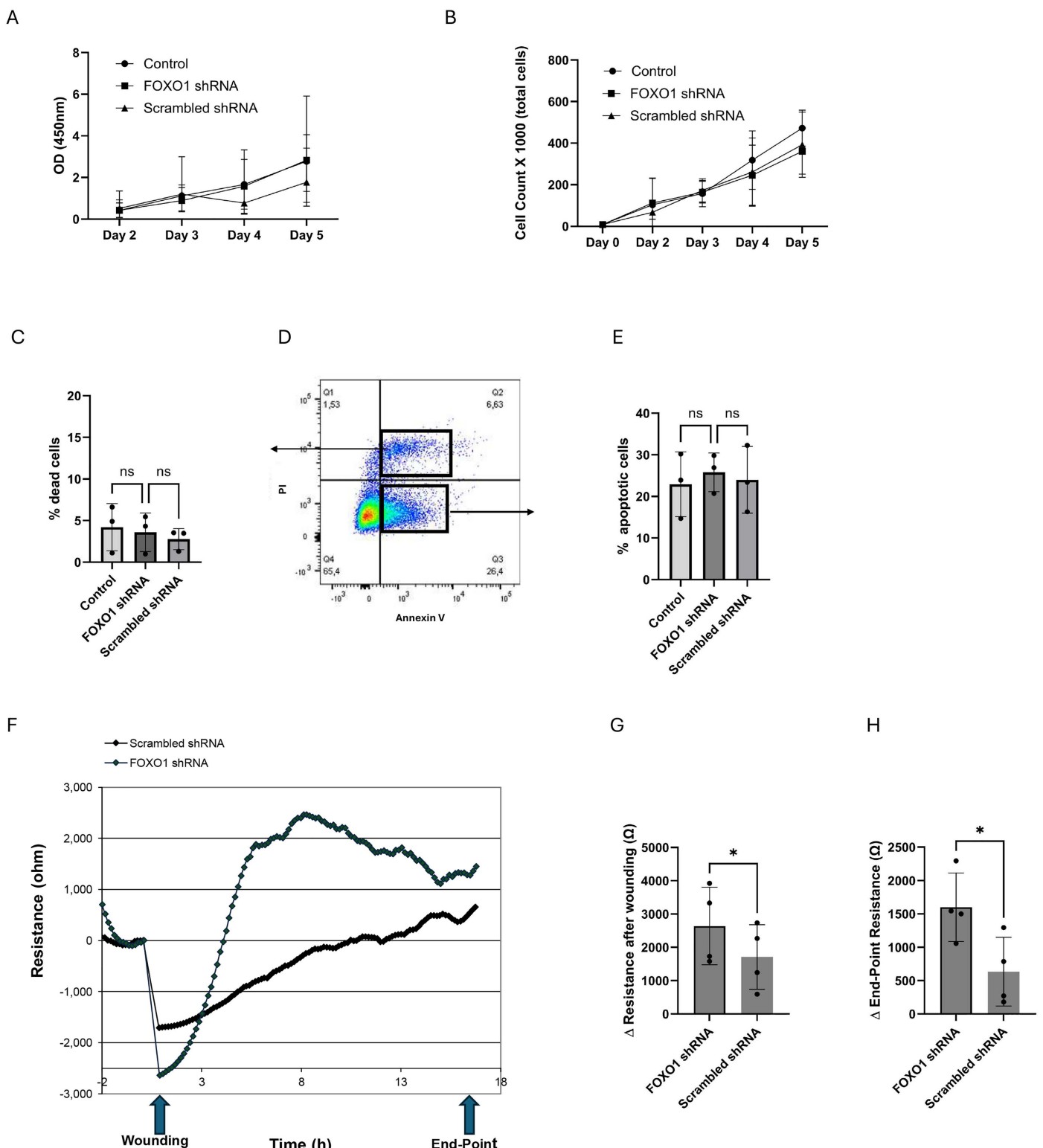

**Fig 2. FOXO1 deficient BEAS-2B cells show no change in cell proliferation and survival but exhibit altered barrier function.** WST-1 proliferation assay (A) and total cell counts (B) for 5 days following plating of BEAS-2B cells treated with FOXO1 shRNA lentivirus, scrambled shRNA lentivirus and untransduced cells (control) (n = 3). Representative flow cytometry plot (D) and percent of dead (C) and apoptotic (E) BEAS-2B cells analyzed

with Annexin V and PI dye (n = 3). Statistical Analysis was performed with ANOVA. F) BEAS-2B monolayer average resistance tracings at 4000 Hz in response to wounding challenge measured with ECIS1600; n = 4 cell replicates per group. Quantification of resistance after wounding (G) and at the end-point (H) shows a significant difference between scrambled shRNA and FOXO1 deficient BEAS-2B cells. Statistical Analysis with t-test* p < 0.05.

### Transcriptional regulation of TLR3 by FOXO1: Insights from in silico and EMSA analysis

Since FOXO1 is a transcription factor, we hypothesized that FOXO1 regulates TLR3 mRNA expression by binding to its promoter. To explore this, we performed *in silico* analysis of publicly available ChIP-seq datasets to assess potential FOXO1–TLR3 interactions. Two relevant datasets derived from Human Umbilical Vein Endothelial Cells (HUVECs) and hepatocellular carcinoma cells (HepG2) were retrieved from the Gene Expression Omnibus [20] and reanalyzed to specifically examine FOXO1 occupancy at the TLR3 locus. ChIP-seq data revealed FOXO1 binding within the promoter region of the TLR3 gene in both cell types, and the canonical FOXO1 binding motif (HOCOMOCO database) was identified in the proximal promoter region in HepG2 cells. Analysis of ChIP-seq datasets from cells expressing constitutively active FOXO1 demonstrated binding signals at the TLR3 locus above background in unstimulated cells, as determined by ChIP-Atlas peak calling. These binding sites were identified through unbiased genome-wide analysis of FOXO1 peaks, rather than through pre-selection of consensus motifs. Together, these findings indicate a basal-level interaction of FOXO1 with the TLR3 locus (Fig 6A).

To experimentally validate our *in silico* findings, we conducted Electrophoretic Mobility Shift Assay (EMSA) using oligos representing the Hep G2 FOXO1 binding site (Fig 6B). Multiple DNA–protein complexes were detected; however, several of these bands were also present in probe-only controls, indicating non-specific binding. While initial assays with lysates from FOXO1-deficient and scrambled shRNA cells showed no specific binding lysates from cells transfected with a CA-FOXO1 plasmid demonstrated clear binding to the oligos (complex I) (Fig 6B). However, there was no increased band intensity (complex I) for the CA-FOXO1 vs the control lysates. Nuclear extracts from BEAS-2B cells transfected with CA-FOXO1 formed complexes I and II upon incubation with the DNA probe, both with and without incubation of FOXO1 monoclonal antibody (mAb), but no supershift was observed (Fig 6C). As no supershift was detected in the presence of FOXO1 antibody, the role of FOXO1 remains inconclusive, as our data do not support direct binding.

Protein extracts from FOXO1-deficient and scrambled shRNA BEAS-2B cells formed DNA–protein complexes I and II, with no detectable differences between the two groups (Fig 6D).

Raw Western blot and EMSA data supporting main figures are listed as supplementary figures (S1A–F Fig).

## Discussion

FOXO1 is widely recognized for its role in immune modulation, apoptosis, and oxidative stress responses [21–23]. Our findings expand on these functions by identifying FOXO1 as a contributor to epithelial responses during viral infections. Across multiple complementary approaches, our data show that FOXO1 modulates TLR3 mRNA expression and influences selected epithelial responses downstream of TLR3 activation, including epithelial recovery from injury. The lack of detectable changes in TLR3 protein levels, despite altered transcript expression levels remain unresolved. Expression and function of immune-related receptors are often decoupled from transcription, as innate immune signalling relies more on post-translational control and adaptor dynamics than on receptor synthesis [24,25].

Our data show that Poly(I:C) stimulation induced a time-dependent increase in FOXO1 nuclear localization, indicating FOXO1 activation downstream of TLR3 activation. We therefore propose that altered outcomes of TLR3 activation in the absence of FOXO1 are most likely the result of lack of TLR3-induced FOXO1 activation and probably not the result of altered TLR3 expression. We also tested whether FOXO1 participates in TLR3-dependent cytokine regulation in airway epithelial cells and found that FOXO1 contributes to the expression of IL-6 and CCL2 but does not affect the release of other cytokines or chemokines, such as GM-CSF, IL8, TSLP, CXCL10, IFN-λ, and TNF-α. This aligns with previous

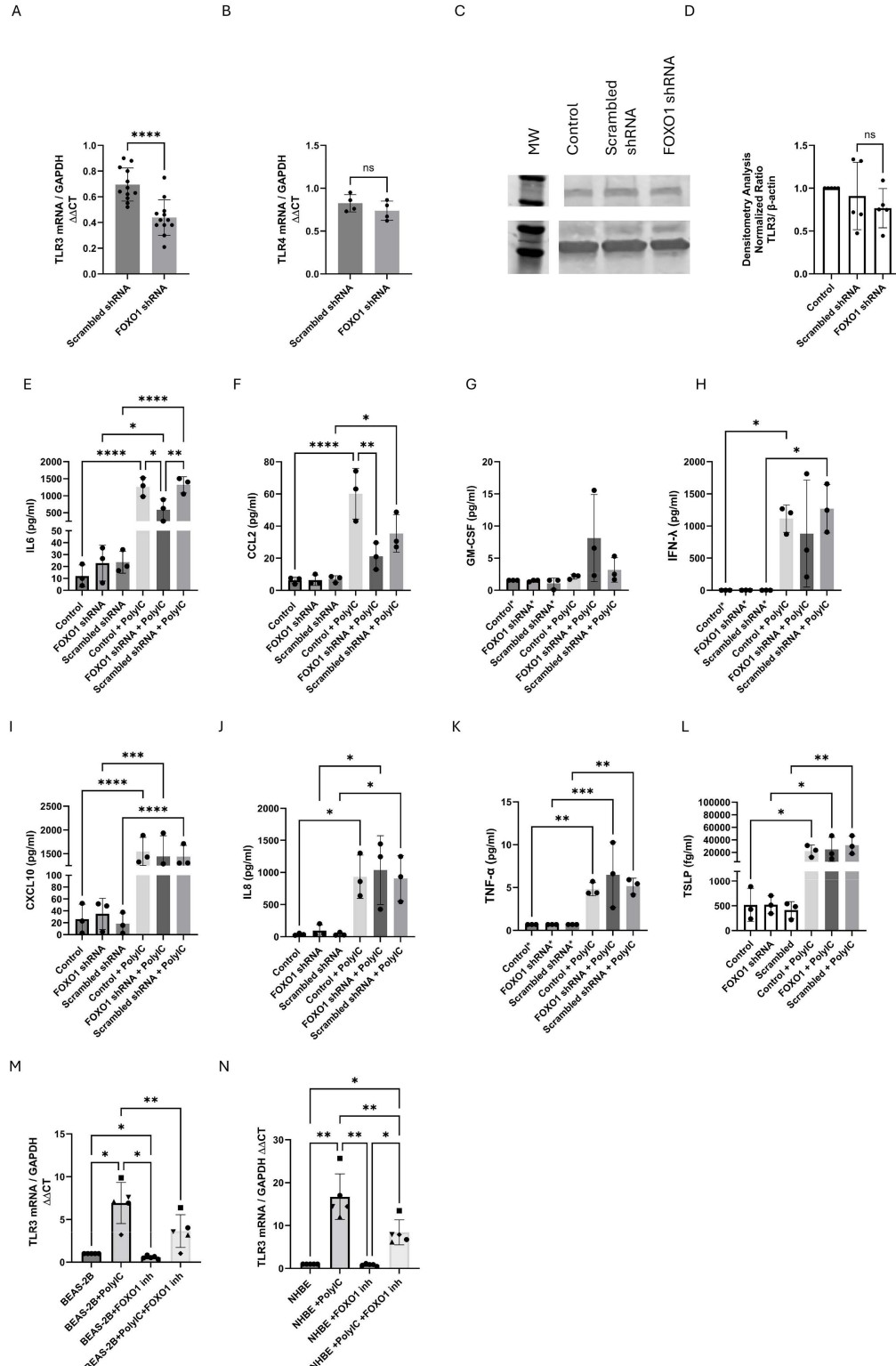

**Fig 3. FOXO1 deficient BEAS-2B cells show reduced TLR3 mRNA expression and reduced cytokine release after TLR3-mediated stimulation.**
A) TLR3 mRNA expression was assessed by RT-qPCR in BEAS-2B cells transduced with FOXO1 shRNA or scrambled shRNA lentivirus. Expression levels were normalized to housekeeping gene GAPDH and expressed relative to untransduced cells (n = 12). Statistical analysis was performed with t-test ****p < 0.0001. B) TLR4 mRNA expression was analyzed by RT-qPCR in BEAS-2B cells transduced with FOXO1 shRNA and scrambled shRNA

lentivirus. Expression was normalized to GAPDH and expressed relative to untransduced cells (n = 4). Statistical analysis with t-test. Representative Western blot (C) and densitometry analysis (D) of TLR3 expression for FOXO1 deficient BEAS-2B cells compared to controls, B-actin was used as a loading control (n = 6). Statistical Analysis with t-test, **p < 0.01. FOXO1 deficient cells and scrambled shRNA cells were stimulated with 50 µg/mL Poly(I:C) for 24 hours and release of IL6 (E),CCL2 (F), GM-CSF (G), IFN-λ (H), CXCL10 (I), IL8 (J), TNF-α (K), and TSLP (L) was tested with an MSD assay (n = 3). TLR3 mRNA expression was assessed by RT-qPCR in BEAS-2B (M) and NHBE (N) cells stimulated with Poly(I:C) for 8 hours in the presence or absence of the FOXO1 inhibitor AS1842856 (1 µM) (n = 5). Statistical analysis was performed using ANOVA *p < 0.05. **p < 0.01, **** P < 0.0001.

studies that demonstrated FOXO1-dependent induction of IL-6 and CCL2 downstream of TLR3 activation in mesenchymal stromal cells [14] and adipocytes [26], supporting a conserved role for FOXO1 in coordinating innate cytokine responses across multiple cell types. This pattern suggests that FOXO1 activity is selective for a subset of inflammatory mediators, while release of other mediators may be maintained through compensatory regulatory mechanisms in airway epithelial cells. This selectivity may depend on the activation stimulus, as FOXO1 activation mediates the release of cytokines/chemokines in response to bacterial infections in airway epithelial cells, but with a different array of cytokines affected [11]. In addition, we showed that TLR3-induced TLR3 mRNA transcription is dependent on FOXO1 activity. Although FOXO1 interacts with transcriptional regulators such as NF-κB [27,28], this alone may not explain the selective effects on cytokine expression we observed. FOXO1's influence on IL-6 and CCL2 may involve cell-specific co-regulatory complexes, chromatin accessibility, or promoter-specific control, suggesting that it modulates rather than broadly suppresses antiviral signaling.

Our findings complement existing evidence that TLR3 transcription is regulated by NF-κB, AP-1, IRF3, and IRF7 [29], and further influenced by post-transcriptional mechanisms such as microRNAs [30]. Notably, FOXO1 itself can be regulated by NF-κB, AP-1, and microRNAs [31,32], underscoring the complexity of this reciprocal network and its potential as a target for modulating antiviral responses.

Despite FOXO1's established role in regulating various aspects of homeostasis, including wound healing, apoptosis, metabolism [10,33–35], its knockdown did not affect cell growth in BEAS-2B cells. This lack of effect may reflect the functional redundancy of other FOXOs, which could sustain homeostatic growth in the absence of FOXO1 [36]. In contrast, studies in keratinocytes showed that FOXO1 deficiency reduced proliferation and impaired wound healing [33], suggesting that the role of FOXO1 in proliferation may be cell type–specific. However, in our experiments we also show altered barrier recovery in ECIS-based wounding assays in FOXO1-deficient airway epithelial cells, indicating that FOXO1 contributes to epithelial barrier integrity and affects wound healing, as it does for keratinocytes. Thus, FOXO1 deficiency alters epithelial barrier properties, suggesting a mechanism by which FOXO1 may influence host–virus interactions at the epithelial surface.

FOXO1 participates in diverse viral response pathways across cell types. In mesenchymal stromal cells, it promotes TLR3-dependent cytokine and migration responses [14], whereas in 293T and THP-1 cells, it suppresses RIG-I signaling by reducing TRAF3 ubiquitination and destabilizing IRF3, thereby limiting type I IFN production [37]. In our experiments, FOXO1 knockdown did not alter transcript levels of RIG-I (DDX58), MAVS, or MYD88 at baseline or after Poly(I:C) stimulation in BEAS-2B airway epithelial cells, suggesting that FOXO1 may not change these pathways at the transcriptional level in the airway epithelium. Given that Poly(I:C) can also engage cytoplasmic sensors such as RIG-I and MDA5, the unchanged expression of other cytokines is consistent with compensatory signaling through these alternative viral sensing pathways [37–39].

In primary airway epithelial cells, pharmacologic FOXO1 inhibition reduced SARS-CoV-2 spike mRNA at 24 hours post-infection, indicating that FOXO1 activity influences viral RNA accumulation. Although FOXO1 has been reported to promote antiviral signaling in some systems [40], our findings support a pathway- and cell-type–specific role for FOXO1 in epithelial immunity. Pharmacologic inhibition of FOXO1 has also been shown to enhance HIV-1 gene expression and replication in resting CD4 T cells, further underscoring the context-dependent effects of FOXO1 on viral infection across

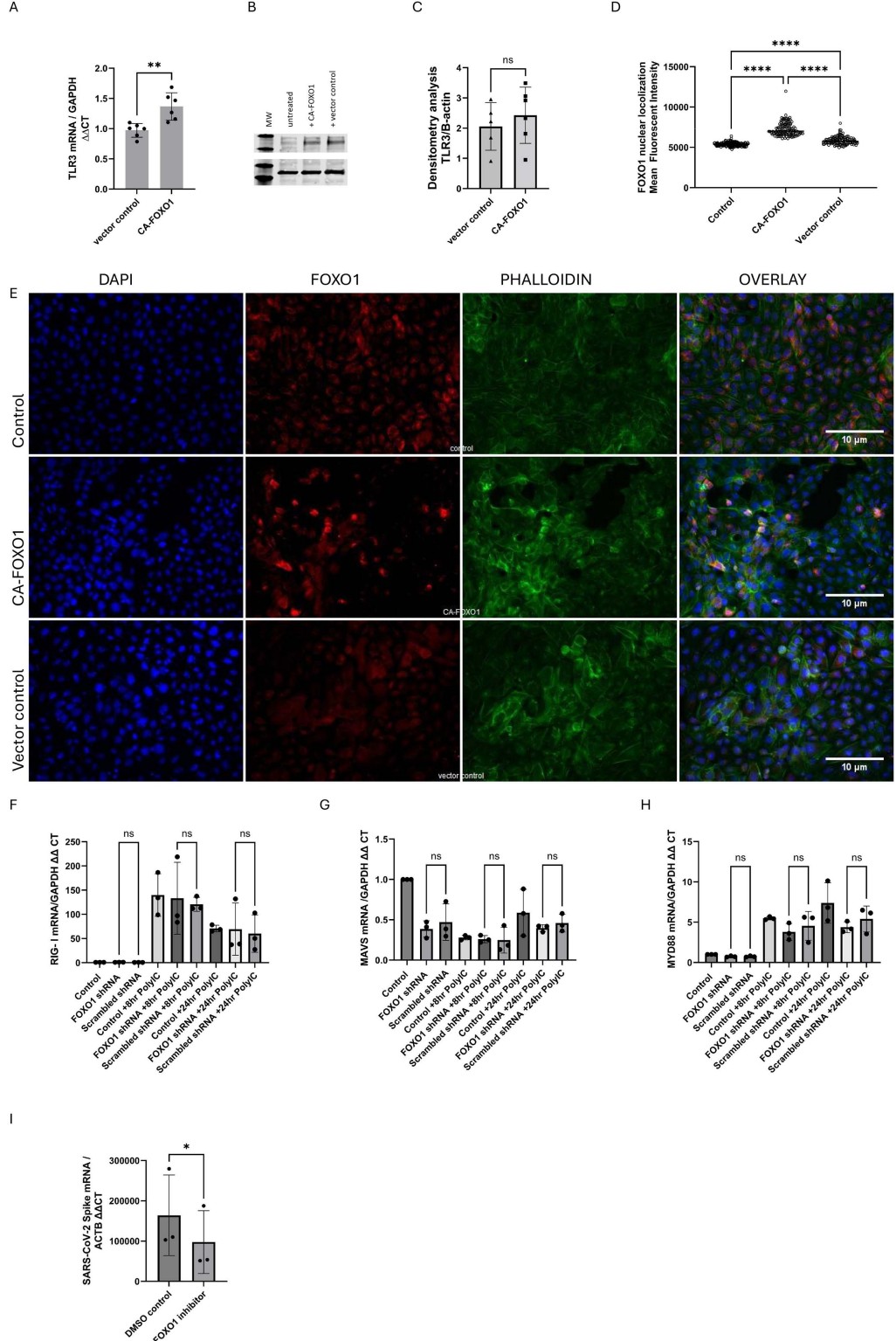

**Fig 4. FOXO1 modulates TLR3 transcription and epithelial responses during viral infection.** A) RT-qPCR showed increased TLR3 mRNA expression for BEAS-2B transfected with a CA-FOXO1 plasmid compared to vector control (cells transfected with an empty plasmid); GAPDH was used as a housekeeping gene (n = 6). Representative Western blot (B) and densitometry analysis (C) of TLR3 expression for BEAS-2B transfected with CA-FOXO1

plasmid compared to vector control, β-actin was used as a loading control (n = 6). Statistical Analysis with t-test, **p < 0.01. D + E) Immunofluorescence staining for BEAS-2B transduced with CA-FOXO1 shows increased FOXO1 protein in the nucleus. FOXO1 (red) was detected using an anti-FOXO1 antibody with a red-fluorescent secondary antibody, F-actin (green) with phalloidin, and nuclei (blue) with DAPI. Images were taken with an Olympus IX81 epifluorescence microscope using a 20X objective lens. Volocity Analysis was used to quantify nuclear localization of FOXO1 by measuring the mean fluorescence intensity of FOXO1 staining colocalized with DAPI. For each group 40−60 cells per slide were analyzed. Statistical Analysis was conducted with ANOVA **** p < 0.001. BEAS-2B cells transduced with FOXO1 or scrambled shRNA lentivirus were analyzed by RT-qPCR for DDX58 (RIG-I, F), MAVS (G), and MYD88 (H) mRNA expression at baseline and after Poly(I:C) stimulation (8 h and 24 h). Expression was normalized to GAPDH and expressed relative to unstimulated scrambled controls (n = 3; ANOVA). (I) NHBE cells were infected with SARS-CoV-2 in the presence or absence of a FOXO1 inhibitor. Total RNA was collected 24 h post-infection, and viral RNA levels were quantified by qRT-PCR, normalized to ACTB, and expressed relative to mock-infected cells (n = 3; paired t-test).

different cell types [41], highlighting that its role in viral pathogenesis depends on the cellular context. Because excessive TLR3 activation can amplify NF-κB–driven inflammation and oxidative or metabolic stress [28–30], FOXO1 could influence how epithelial cells balance responses to viral infection with inflammatory control.

While BEAS-2B cells are widely used as a model for airway epithelial function, their transformed nature may reduce FOXO1's influence, as oncogenic pathways can override FOXO1-mediated growth regulation. Initial observations in NHBE cells treated with the FOXO1 inhibitor show reduced TLR3 expression, suggesting that FOXO1-dependent regulation of TLR3 may also be present in NHBE cells. Future studies using primary bronchial epithelial cells or in vivo models will help further define the physiological relevance of FOXO1-dependent regulation in the airway epithelium.

In silico analysis provides a starting point for identifying potential FOXO1–TLR3 interactions, but experimental validation is essential for determining their functional relevance. Our EMSA results indicate that while nuclear proteins can bind the predicted FOXO1 site in the TLR3 promoter, direct FOXO1 involvement remains unconfirmed. Several bands overlapped with probe-only controls, and complex II did not supershift with FOXO1 antibody, underscoring the limitations of EMSA outside the native chromatin context. These findings suggest that any regulatory effects of FOXO1 on TLR3 are likely indirect or dependent on additional cofactors or signalling pathways.

In summary, our findings identify FOXO1 as a stimulus-responsive transcriptional regulator that contributes selectively to airway epithelial responses following viral RNA sensing. FOXO1 modulates TLR3 mRNA expression and is dynamically engaged upon TLR3 activation but does not broadly alter downstream antiviral signaling or baseline TLR3 protein abundance. Instead, FOXO1 influences a restricted set of epithelial outputs, including selective cytokine production and epithelial barrier recovery, and affects epithelial responses during SARS-CoV-2 infection. Together, these results support a model in which FOXO1 orchestrates specific epithelial antiviral and inflammatory responses downstream of viral sensing, rather than functioning as a global regulator of innate immune signaling pathways in the airway epithelium.

## Supporting information

**S1 Fig. Raw Western blot and EMSA data supporting main figures.** A) Raw Western blot data corresponding to Fig 1B. B) Raw Western blot data corresponding to Fig 3C. C) Raw Western blot data corresponding to Fig 4B. D) Raw EMSA data corresponding to Fig 5B. E) Raw EMSA data corresponding to Fig 5C. F) Raw EMSA data corresponding to Fig 5D. G) FOXO1 inhibitor decreases TLR3 mRNA expression in BEAS-2B cell in a dose-dependent fashion. BEAS-2B cells were treated with Poly(I:C) alone or with increasing concentrations of a FOXO1 inhibitor (0.01μM, 0.1 μM and 1.0 μM). Untreated cells and cells treated with Poly(I:C) alone served as controls. TLR3 mRNA expression was quantified by RT-qPCR and normalized to GAPDH. BEAS-2B cells showed reduced TLR3 expression at baseline compared to cells treated with FOXO1 inhibitor AS1842856. BEAS-2B in the presence of Poly(I:C) and FOXO1 inhibitor AS1842856 showed a dose-dependent reduction in TLR3 mRNA expression (n = 3).
(PDF)

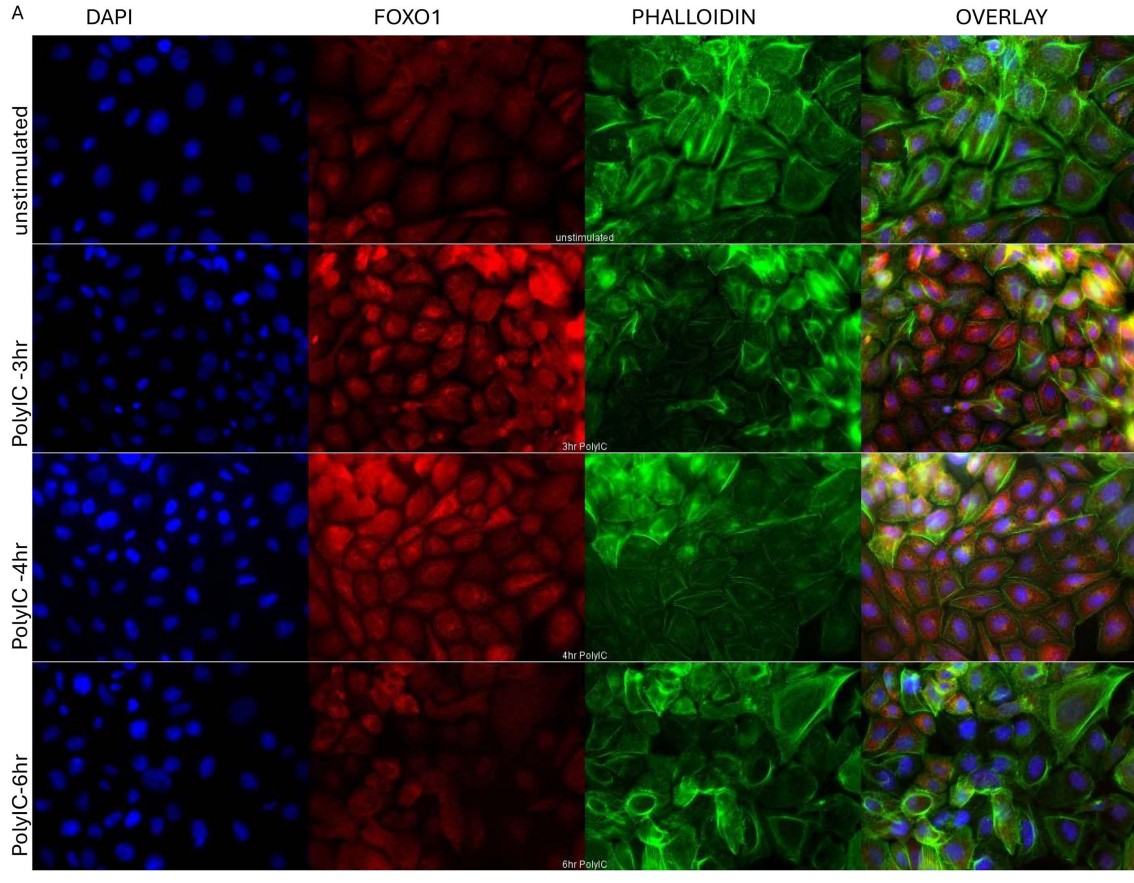

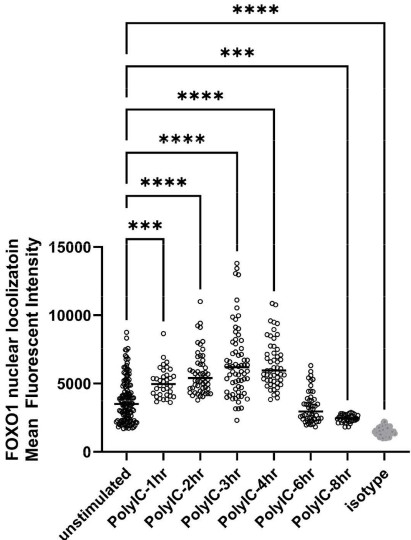

**Fig 5. FOXO1 nuclear fluorescence intensity increases following Poly(I:C) stimulation in BEAS-2B airway epithelial cells.** A) Representative set of Immunofluorescence staining for BEAS-2B cells stimulated with 50 µg/mL Poly(I:C) at different time points. FOXO1 (red) was detected using an anti-FOXO1 monoclonal antibody, nuclei were stained with DAPI (blue), and the cytoskeleton was visualized with phalloidin (green). Images were captured

at 60× magnification using a DeltaVision Confocal Microscope. B) Mean fluorescence intensity of FOXO1 localized to the nucleus was quantified with Volocity software. For each group 40–60 cells were analyzed. Statistical Analysis was performed using Anova ***P < 0.001, **** P < 0.0001.

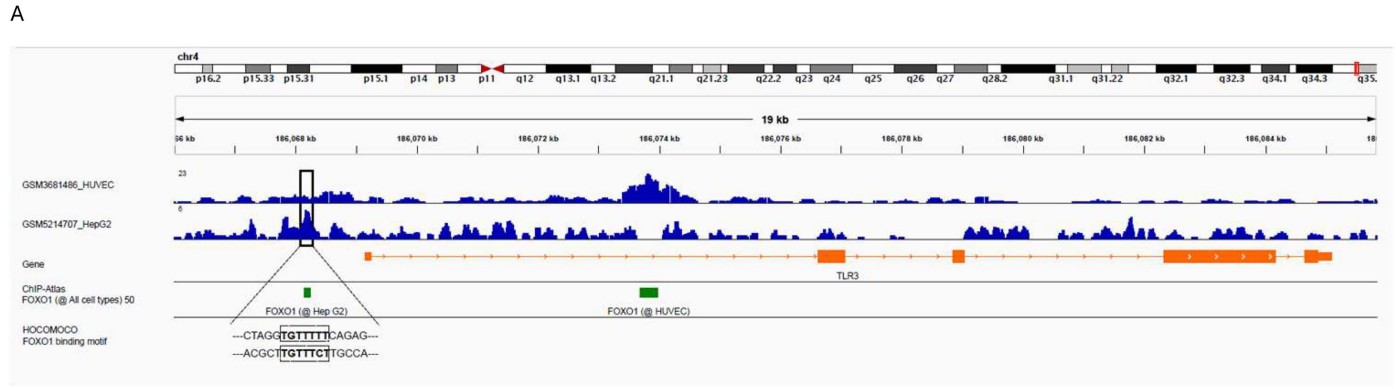

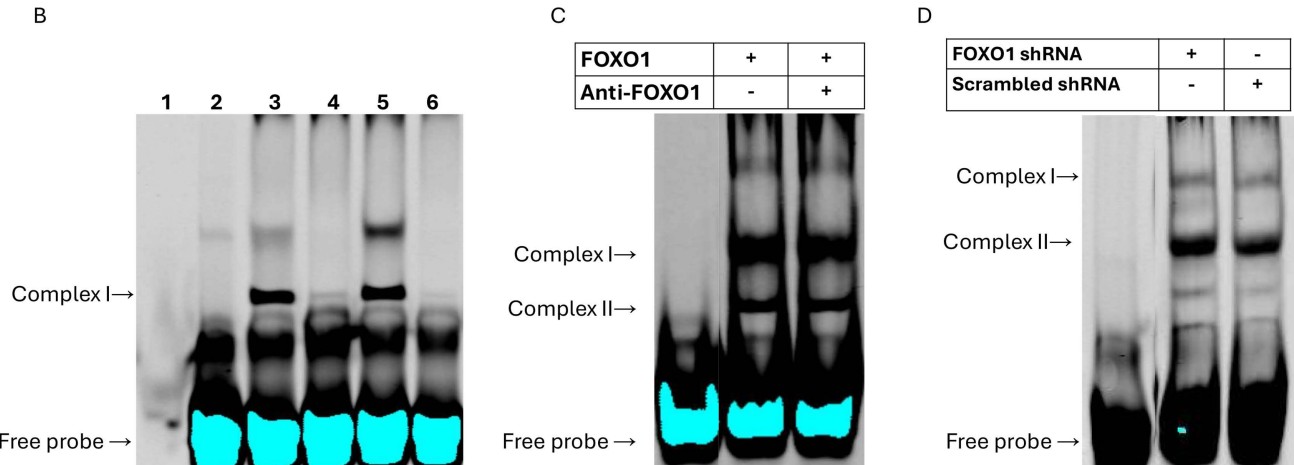

**Fig 6. Predicted FOXO1 binding motifs in the TLR3 promoter are not supported by EMSA evidence of direct binding.** A) The top panel presents FOXO1 ChIP-Seq peaks retrieved from Gene Expression Omnibus (GSM3681486) in HUVEC cells and (GSM5214707) in the HepG2 cell line. The bottom panel shows FOXO1 binding sites from the ChIP-Atlas visualized with IGV (Integrative Genomics Viewer). The FOXO1 motif from the HOCOMOCO database within the proximal promoter sequence of the TLR3 gene is highlighted below. B) EMSA with nuclear extracts from BEAS-2B cells incubated with FOXO1-TLR3 Promoter Oligos. Nuclear extracts from BEAS-2B cells transfected with CA-FOXO1 or plasmid control. Lane 1: dye only, Lane 2: probe only, Lane 3: nuclear extracts from CA-FOXO1 BEAS-2B cells, Lane 4: nuclear extracts from CA-FOXO1 BEAS-2B cells + cold competitor, Lane 5: nuclear extracts from vector control BEAS-2B cells, Lane 6: nuclear extracts from vector control BEAS-2B cells + cold competitor. Lane 3 shows complex I formation, which disappears in lane 4, indicating non-specific binding. C) Nuclear extracts of BEAS-2B cells transfected with CA-FOXO1 incubated with or without FOXO1 mAb shows the formation of complex I + II, but no supershift occurred. D) Incubation of protein extracts from BEAS-2B FOXO1 deficient and scrambled control lines show formation of complexes I and II but no difference is observed between cell lines.

## Acknowledgments

We would liketo acknowledge Dr. Fred Berry for his help with the EMSA, and Luke Gerla and Marc Duchenne for their help with cytokine measurements.

## Author contributions

**Conceptualization:** Nadia M. Daniel, Harissios Vliagoftis, Joaquin López-Orozco.

**Data curation:** Nadia M. Daniel.

**Formal analysis:** Nadia M. Daniel, Ritu Mann-Nüttel.

**Funding acquisition:** Harissios Vliagoftis, Paul Forsythe.

**Investigation:** Nadia M. Daniel, Joaquin López-Orozco.

**Methodology:** Nadia M. Daniel, Nami Shrestha Palikhe.

**Software:** Ritu Mann-Nüttel.

**Supervision:** Harissios Vliagoftis, Paul Forsythe, Tom Hobman.

**Writing – original draft:** Nadia M. Daniel.

**Writing – review & editing:** Harissios Vliagoftis, Paul Forsythe, Nami Shrestha Palikhe, Ritu Mann-Nüttel, Joaquin López-Orozco.

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
