## [Decision Letter · Decision Letter 0]

16 Aug 2025

Dear Dr. Daniel,

Thank you for submitting your manuscript to PLOS ONE. After careful consideration, we feel that it has merit but does not fully meet PLOS ONE’s publication criteria as it currently stands. Therefore, we invite you to submit a revised version of the manuscript that addresses the points raised during the review process.

Thank you for submitting the following manuscript to PLOS ONE.

Please revise the manuscript according to the reviewers' comments and upload the revised file.

We look forward to receiving your revised manuscript.

Kind regards,

Yung-Hsiang Chen, Ph.D.

Academic Editor

PLOS ONE

Journal Requirements:

3. Please expand the acronym “NSERC” (as indicated in your financial disclosure) so that it states the name of your funders in full.

This work was supported by a Discovery grant from NSERC and the GSK/CIHR Chair in Airway Inflammation to HV. NMD was supported by an NSERC Canada Graduate Scholarship Doctoral Program studentship. PF is the AstraZeneca (Canada) Inc., Chair in Asthma and Obstructive Lung Disease.

This work was supported by a Discovery grant from NSERC and the GSK/CIHR Chair in Airway Inflammation to HV. NMD was supported by an NSERC Canada Graduate Scholarship Doctoral Program studentship. PF is the AstraZeneca (Canada) Inc., Chair in Asthma and Obstructive Lung Disease. We would want to acknowledge Dr. Fred Berry for his help with the EMSA, and Luke Gerla and Marc Duchenne for their help with cytokine measurements.

This work was supported by a Discovery grant from NSERC and the GSK/CIHR Chair in Airway Inflammation to HV. NMD was supported by an NSERC Canada Graduate Scholarship Doctoral Program studentship. PF is the AstraZeneca (Canada) Inc., Chair in Asthma and Obstructive Lung Disease.

6. We notice that your supplementary figures are uploaded with the file type 'Figure'. Please amend the file type to 'Supporting Information'. Please ensure that each Supporting Information file has a legend listed in the manuscript after the references list.

Additional Editor Comments:

Thank you for submitting the following manuscript to PLOS ONE.

Please revise the manuscript according to the reviewers' comments and upload the revised file.

Reviewers' comments:

Reviewer's Responses to Questions

**Comments to the Author**

1. Is the manuscript technically sound, and do the data support the conclusions?

Reviewer #1: No

Reviewer #2: Partly

2. Has the statistical analysis been performed appropriately and rigorously?

Reviewer #1: Yes

Reviewer #2: Yes

3. Have the authors made all data underlying the findings in their manuscript fully available?

Reviewer #1: Yes

Reviewer #2: Yes

4. Is the manuscript presented in an intelligible fashion and written in standard English?

Reviewer #1: Yes

Reviewer #2: No

Reviewer #1: Review Comments

In this manuscript, the authors report that FOXO1 regulates TLR3 expression in airway epithelial cells, and that such regulation secondarily controls the induction of TLR3-dependent cytokine expression. The study addresses the involvement of FOXO1 in antiviral innate immune responses. While some of the findings are of interest, the work remains preliminary, and additional experiments are required before it can be considered for publication. The following points summarize the reviewer’s concerns:

1. The authors claim that FOXO1 regulation of TLR3 in airway epithelial cells is important for inflammatory cytokine production in antiviral innate immunity. However, TLR3 was evaluated only at the mRNA level (Figure 3). Although the changes may be statistically significant, the biological impact is unclear. It is essential to examine whether TLR3 protein levels are also reduced, as mRNA changes do not necessarily correspond to protein changes. Since TLR3 immunoblotting is shown in Figure 4, such experiments should be feasible.

2. While the authors discuss the importance of TLR3 in antiviral innate immune responses in airway epithelial cells, it is well established that in non-professional immune cells, viral RNA sensing is primarily mediated by RIG-like receptors such as RIG-I, whereas TLR3 serves as a major dsRNA sensor in professional immune cells like dendritic cells. The study does not address the contribution of RIG-I at all. Although the focus on TLR3 is understandable, the authors should also investigate the effect of FOXO1 knockdown/overexpression on RIG-I expression and signaling.

3. The regulation of TLR3 expression by FOXO1 may be a secondary effect; the role of FOXO1 downstream of RIG-I–MAVS or TLR3–MyD88 signaling may be more important. Prior studies (PMID: 30944148) have shown that FOXO1 is involved in controlling TLR3-dependent cytokine expression. There are also reports on FOXO1 regulation in RIG-I signaling (PMID: 33187908). If the authors aim to emphasize the importance of FOXO1-mediated regulation of TLR3, they should also present data on FOXO1 function in TLR3 and RIG-I pathways.

4. On line 300, the authors state that FOXO1 is essential for the constitutive expression of TLR3. However, well-known TLR3-dependent genes such as IP-10 and IL-8, which have been reported to be induced by poly(I:C), show no changes according to Figure S1. This is a major inconsistency that requires clarification.

5. While FOXO1 appears to suppress IL-6 and CCL2 expression, the effect is limited, and for IL-6 it explains only part of the observed changes. This point requires further discussion or additional experimental validation.

6. The current data are based on airway epithelial cells. Is the observed FOXO1 function specific to airway epithelial cells, or is it a general mechanism across cell types? This point should be discussed.

7. Since this is presented as a study of antiviral responses in airway epithelial cells, at least one set of data from an actual viral infection experiment should be provided.

Minor Points

1. Figure 1F is too dark to clearly visualize FOXO1-positive staining; a brighter image should be provided.

2. Figure S1 presents important findings and should be moved from the supplemental data to the main figures.

3. Likewise, the results in Figure S4 should be presented as a main figure rather than in the supplemental section.

Reviewer #2: Recommendation: Major Revision

Review question: Is the manuscript technically sound, and do the data support the conclusions?

Answer: Party

General statement: The paper provides some evidence that FOXO1 may either directly or indirectly regulate TLR3 expression in BEAS2B cells, both unstimulated, and stimulated with PolyIC, but some of the pieces of evidence are incomplete. Additionally, there is no/little evidence to say whether this interaction is direct or indirect (ie. no convincing evidence to say that FOXO1 binds the TLR3 locus in BEAS2B cells, and no evidence that it does not). Finally, is not clear if any evidence is provided for FOXO1 regulating TLR3 signalling per say, only FOXO1 impacting cytokine expression and barrier function, that may be dependent or independent of TLR3.

Specific questions/suggestions:

1) Please justify and confirm use of 50ug/ul Poly (I:C) concentration for TLR3 agonism (line 246 states “based on these findings” 50ug/mL was used, but only one concentration was tested, and this was different from the methods where 50ug/ul was stated).

2) It is not clear in the text how FOXO1 is suggested to become active and translocate to the nucleus, which would justify why the constitutively active FOXO1 mutant (CA-FOXO1) was used. It seems that this mutant is phosphorylation-deficient, meaning phosphorylation inhibits its activity, so please describe this in the text.

3) In the methods, please specify that nuclear extracts were used for EMSA and how these were prepared, in S4 it is stated nuclear extracts were used but this was not in the methods. Please also clarify all the components that were used in the “EMSA buffer”, or if this was part of commercial kit.

4) (Figure 1) For the FOXO1 shRNA transduced cells, the western blot shows overall less FOXO1

compared to control, but this is much less clear in the immunofluorescence data where only a drop in nuclear localization is found. Please show whole cell fluorescent intensity for FOXO1 as well.

5) (Figure 1) It would be useful to show reduction in TLR3 protein (ie. through western blot since you have a working antibody) as well as the mRNA in Figure 1 following FOXO1 shRNA, since this links the regulatory element of your story to the functional statements in Figure 2.

6) (Figure 4) It is not clear what the significance of TLR3 regulation by FOXO1 is, if TLR3 expression is not impacted at the protein level, and how this contributed to the functional differences you observe? Since it is stated that (potentially) a different timepoint could be used to capture protein changes, this should be done tested, or ensure that the antibody is detecting the correct target. The same goes for figure 3 with the FOXO1 inhibitor.

7) (Figure 5) Please describe whether these ChIP datasets used are treated or use constitutively active FOXO1. It is not clear whether there is any major binding above the background signal at the two FOXO1 motifs selected, likely because there is not much FOXO1 binding to the TLR3 locus without simulation. Therefore, this alone can not be used as evidence that FOXO1 can bind the TLR3 locus in any cell type. It is suggested to perform a ChIP-PCR in Poly-IC stimulated conditions (where there would be greater binding compared to background) or find a different data set that is more convincing.

8) (Figure S5) Although there is evidence of some binding of proteins within the nuclear extract to the oligo in the EMSA based on lane 2 compared to lane 3 (panel A, labelled “complex 2”), all other labelled complexes are present in the probe only lane so these should not be discussed as binding in panel B+C. Additionally, this probe only control should also be present in panels B+C. Since “complex 2” does not super-shift with FOXO1 antibody, there is no evidence that FOXO1 binds this oligo, even in constitutively active conditions. There is quite a lot of DNA on either end of the 4bp FOXO1 binding site, so it could be a host of other factors binding to this oligo, and this is not in the native chromatin conformation or cell environment that ChIP-PCR would allow.

9) The “signalling” part of the title should be removed unless additionally evidence is provided that TLR3 signalling, ie. through phospho-IRF3, phospho-IKKε, phospho-TBK1, or ISG activity is impacted by FOXO1 depletion/inhibition (and that TLR3 protein is reduced by FOXO1 depletion as stated above). Instead focus on the barrier function and cytokine expression impacts of FOXO1.

Review Question: Is the manuscript presented in an intelligible fashion and written in standard English?

Answer: Party (only yes or no option available)

Story is overall intelligible, but some changes are suggested. Some grammatical errors are not noted here so please review text before resubmitting.

Line 206 grammar “of by total cell counts at the end of incubation”

Line 206 “Barrier integrity showed that both FOXO1-deficient and scrambled shRNA”, explain how resistance relates to barrier integrity ie. higher ohms tighter barrier: barrier integrity did not “show”, higher olms indicated increased barrier integrity

Line 228 “FOXO1 downregulation did not affect baseline release of cytokines from BEAS-2B cells” unknown whether this is all cytokines, only chosen/select cytokines

Line 242 Specify activation is by poly-IC in paper cited, would help with overall story

Line 281 Please indicate whether this data is or is not shown

.

Reviewer #1: **Yes:**Tomoh MatsumiyaTomoh MatsumiyaTomoh MatsumiyaTomoh Matsumiya

Reviewer #2: No

---

## [Author Response · Author response to Decision Letter 1]

1 Dec 2025

Date: 24 November 2025

Manuscript Number: PONE-D-25-41597] - [EMID:191f4936a3b0eb60]

Title of Article: FOXO1 transcription factor regulates Toll-like-receptor 3 mRNA and coordinates antiviral responses in BEAS-2B airway epithelial cells.

Name of the Corresponding Author: Nadia Daniel

Email Address of the Corresponding Author: ndaniel@ualberta.ca

Dear Editors,

We thank you for your insightful feedback and comments, and for giving us the opportunity to improve our manuscript.

We have carefully considered each of the comments and made the appropriate changes to our manuscript. We summarize our changes as follows:

• We added additional data and 2 co-authors

• We updated the methodology and the discussion has been extensively reorganized

• We refined the title and text to more accurately reflect the scope of our findings.

We also provide detailed responses to each of the comments and the lines in the manuscript that were revised.

We are happy to answer any further questions and look forward to hearing from you.

Yours sincerely,

Nadia Daniel and Harissios Vliagoftis

Editor Comments:

Comment E1. Please ensure that your manuscript meets PLOS ONE's style requirements, including those for file naming. The PLOS ONE style templates can be found at

Response E1: The manuscript and associated files have been revised to comply with the PLOS ONE style requirements. We have carefully cross-checked the submission against the PLOS ONE formatting templates provided and confirm that the revised version meets these requirements.

Comment E2. PLOS ONE now requires that authors provide the original uncropped and unadjusted images underlying all blot or gel results reported in a submission’s figures or Supporting Information files. This policy and the journal’s other requirements for blot/gel reporting and figure preparation are described in detail at https://journals.plos.org/plosone/s/figures#loc-blot-and-gel-reporting-requirements and https://journals.plos.org/plosone/s/figures#loc-preparing-figures-from-image-files. When you submit your revised manuscript, please ensure that your figures adhere fully to these guidelines and provide the original underlying images for all blot or gel data reported in your submission. See the following link for instructions on providing the original image data: https://journals.plos.org/plosone/s/figures#loc-original-images-for-blots-and-gels.

Response E2: We have provided the original, uncropped, and unadjusted blot images corresponding to all Western blot and EMSA data presented in the manuscript. These raw image files are included in the revised submission as Supporting Information. Each file is labelled to indicate the corresponding figure panel in the main text. All blots included in the manuscript are available, and no raw blot/gel images are missing. In our cover letter, we have confirmed that the original blot image data are provided in the Supporting Information.

Comment E3. Please expand the acronym “NSERC” (as indicated in your financial disclosure) so that it states the name of your funders in full. This information should be included in your cover letter; we will change the online submission form on your behalf.

Response E3: In our financial disclosure, we have expanded the acronym “NSERC” to its full form: Natural Sciences and Engineering Research Council of Canada (NSERC). This information is also included in the cover letter for clarity.

Comment E4. Thank you for stating the following financial disclosure: “This work was supported by a Discovery grant from NSERC and the GSK/CIHR Chair in Airway Inflammation to HV. NMD was supported by an NSERC Canada Graduate Scholarship Doctoral Program studentship. PF is the AstraZeneca (Canada) Inc., Chair in Asthma and Obstructive Lung Disease.” Please state what role the funders took in the study. If the funders had no role, please state: "The funders had no role in study design, data collection and analysis, decision to publish, or preparation of the manuscript." If this statement is not correct you must amend it as needed. Please include this amended Role of Funder statement in your cover letter; we will change the online submission form on your behalf.

Response E4: The funders had no role in study design, data collection and analysis, decision to publish, or preparation of the manuscript. We have added this statement to the cover letter as requested.

Comment E5. Thank you for stating the following in the Acknowledgments Section of your manuscript: “This work was supported by a Discovery grant from NSERC and the GSK/CIHR Chair in Airway Inflammation to HV. NMD was supported by an NSERC Canada Graduate Scholarship Doctoral Program studentship. PF is the AstraZeneca (Canada) Inc., Chair in Asthma and Obstructive Lung Disease. We would want to acknowledge Dr. Fred Berry for his help with the EMSA, and Luke Gerla and Marc Duchenne for their help with cytokine measurements.”

This work was supported by a Discovery grant from NSERC and the GSK/CIHR Chair in Airway Inflammation to HV. NMD was supported by an NSERC Canada Graduate Scholarship Doctoral Program studentship. PF is the AstraZeneca (Canada) Inc., Chair in Asthma and Obstructive Lung Disease. Please include your amended statements within your cover letter; we will change the online submission form on your behalf.

Response to Comment E5: All funding-related text has been removed from the Acknowledgments section of the manuscript, leaving only acknowledgment of individual contributions. The corrected Funding Statement is provided in the cover letter, as requested:

Comment E6. We notice that your supplementary figures are uploaded with the file type 'Figure'. Please amend the file type to 'Supporting Information'. Please ensure that each Supporting Information file has a legend listed in the manuscript after the references list.

Response E6: All supplementary figure files have been re-uploaded under the file type “Supporting Information,” and each is now provided with a corresponding legend listed in the manuscript after the References section, as required.

Comment E7. If the reviewer comments include a recommendation to cite specific previously published works, please review and evaluate these publications to determine whether they are relevant and should be cited. There is no requirement to cite these works unless the editor has indicated otherwise.

Response E7: We carefully reviewed all of the publications suggested by the reviewers and have cited those we deemed relevant and appropriate in the revised manuscript.

Comment R1.1: The authors claim that FOXO1 regulation of TLR3 in airway epithelial cells is important for inflammatory cytokine production in antiviral innate immunity. However, TLR3 was evaluated only at the mRNA level (Figure 3). Although the changes may be statistically significant, the biological impact is unclear. It is essential to examine whether TLR3 protein levels are also reduced, as mRNA changes do not necessarily correspond to protein changes. Since TLR3 immunoblotting is shown in Figure 4, such experiments should be feasible.

Response R1.1: We performed western blot analysis of TLR3 protein following FOXO1 knockdown. These experiments showed no statistically significant differences in TLR3 protein expression compared with scrambled controls, despite the consistent reduction at the mRNA level. We have now included these data in the revised manuscript (new Figure 3C&D). We agree that reduced TLR3 mRNA without detectable protein change may limit the functional impact of our observation, and we have expanded on this point in the revised Discussion (lines 428-431).

Comment R1.2: While the authors discuss the importance of TLR3 in antiviral innate immune responses in airway epithelial cells, it is well established that in non-professional immune cells, viral RNA sensing is primarily mediated by RIG-like receptors such as RIG-I, whereas TLR3 serves as a major dsRNA sensor in professional immune cells like dendritic cells. The study does not address the contribution of RIG-I at all. Although the focus on TLR3 is understandable, the authors should also investigate the effect of FOXO1 knockdown/overexpression on RIG-I expression and signaling.

Comment R1.3: The regulation of TLR3 expression by FOXO1 may be a secondary effect; the role of FOXO1 downstream of RIG-I–MAVS or TLR3–MyD88 signaling may be more important. Prior studies (PMID: 30944148) have shown that FOXO1 is involved in controlling TLR3-dependent cytokine expression. There are also reports on FOXO1 regulation in RIG-I signaling (PMID: 33187908). If the authors aim to emphasize the importance of FOXO1-mediated regulation of TLR3, they should also present data on FOXO1 function in TLR3 and RIG-I pathways.

Response R1.2 & R1.3: We agree with the reviewers that other antiviral pathways are also very important in airway epithelial cells. To this effect, we performed qPCR analyses of DDX58 (RIG-I) and its signaling adaptor MAVS, as well as MYD88, in FOXO1 knockdown BEAS-2B cells at baseline and after poly(I:C) stimulation (8 h and 24 h). We observed no significant changes in expression levels of RIG-I, MAVS, or MYD88 between FOXO1 knockdown and control cells under these conditions.

We recognize that prior studies have reported roles for FOXO1 in antiviral signaling. PMID: 30944148 showed that in human mesenchymal stromal cells, FOXO1 regulates TLR3-dependent cytokine and migration responses, whereas PMID: 33187908 demonstrated that FOXO1 negatively regulates RIG-I signaling by limiting TRAF3 ubiquitination and IRF3 stability, thereby reducing type I IFN production. In airway epithelial cells, FOXO1 knockdown decreases TLR3 mRNA but not protein, so we do not infer regulation of receptor abundance. Instead, our findings suggest that FOXO1 primarily modulates downstream signaling competence in response to viral RNA sensors, consistent with its signaling roles reported in other systems. This likely reflects pathway-dependent regulation, where FOXO1 influences antiviral signaling outputs according to the dominant RNA-sensing mechanisms active in each cell type.

We have added these data as Figure 4F+G+H in the revised manuscript. We agree that the absence of changes in RIG-I, MAVS, or MYD88 suggests that FOXO1’s regulatory effects are selective rather than global, and we have revised the Discussion to reflect this specificity (lines 468-473).

Comment R1.4: On line 300, the authors state that FOXO1 is essential for the constitutive expression of TLR3. However, well-known TLR3-dependent genes such as IP-10 and IL-8, which have been reported to be induced by poly(I:C), show no changes according to Figure S1. This is a major inconsistency that requires clarification.

Response R1.4: As we also mentioned above, although FOXO1 knockdown consistently decreased TLR3 transcript levels, this was not accompanied by detectable differences in TLR3 protein or in the induction of these canonical TLR3-dependent cytokines. We have therefore revised the text to remove the statement that FOXO1 is “essential” for constitutive TLR3 expression and now state that FOXO1 influences TLR3 transcription, but the biological significance of this change remains unresolved (lines 428-429). This probably explains why both baseline and poly(I:C)-activated IP-10 and IL8 expression is not altered in FOXO1 deficient cells.

Comment R1.5: While FOXO1 appears to suppress IL-6 and CCL2 expression, the effect is limited, and for IL-6 it explains only part of the observed changes. This point requires further discussion or additional experimental validation.

Response R1.5: We agree that FOXO1’s regulatory effect on IL6 and CCL2 is partial. These results are, however, consistent with previous reports for FOXO1-dependent regulation of these two cytokines by TLR3 in mesenchymal stromal cells and adipocytes (Kim et al.,2019 and Ito et al.,2009). In addition, both IL6 and CCL2 are well recognized to be under the control of multiple transcriptional regulators, including NF-κB, MAPKs, STATs, and IRFs, which act in parallel or redundant fashion to amplify inflammatory outputs. Our findings therefore suggest that FOXO1 functions as a modulator rather than a sole regulator of these cytokines in airway epithelial cells. We have revised the Discussion (lines 433-443) to highlight this point and to caution against over-attribution of IL6 and CCL2 regulation to FOXO1 alone.

Comment R1.6: The current data are based on airway epithelial cells. Is the observed FOXO1 function specific to airway epithelial cells, or is it a general mechanism across cell types? This point should be discussed.

Response R1.6: Our study is focused on airway epithelial cells given their critical role in frontline antiviral defense and the particular interest of our laboratory to lung immunity. However, we agree it is important to present our data in the context of the extensive regulatory role of FOXO1 in other cell types and other tissues. Prior literature demonstrates that FOXO1 participates in immune and inflammatory regulation across diverse systems. For example, FOXO1 regulates macrophage polarization in allergic inflammation (Chung et al., 2016), promotes IRF3 degradation to dampen antiviral responses (Lei et al., 2013), and controls IL6 and CCL2 induction downstream of TLR3 in mesenchymal stromal cells (Kim et al., 2019). FOXO1 has also been shown to drive proinflammatory gene expression in adipocytes (Ito et al., 2009) and to enhance chemokine-mediated lymphocyte recruitment (Miao et al., 2012). More recently, microRNA-dependent regulation of FOXO1 was linked to inflammatory responses in macrophages (Hu et al., 2022). We have expanded the discussion (lines 456-464) to include the studies cited above to help the reader appreciate the wider context of the FOXO1 effects on the immune system.

Comment R1.7: Since this is presented as a study of antiviral responses in airway epithelial cells, at least one set of data from an actual viral infection experiment should be provided.

Response R1.7: We agree with the reviewer that data on the role of FOXO1 in viral infections of airway epithelial cells would strengthen our findings. To address this, we performed an infection experiment in primary human airway epithelial cells (NHBE). Cells were infected with SARS-CoV-2 VIDO-01 (MOI = 0.5) in the presence or absence of a FOXO1 inhibitor, and total RNA was harvested at 24 h post-infection. Viral RNA levels were quantified by qRT-PCR (spike mRNA normalized to β-actin; n = 3; paired t-test). FOXO1 inhibition significantly reduced spike mRNA relative to DMSO controls (Figure 4I). These results and discussion (lines 350-353 & 474-484) are now updated in the revised manuscript and further highlight the context-dependent role of FOXO1 in modulating TLR3-dependent signaling in airway epithelial cells.

Minor Point 1: Figure 1F is too dark to clearly visualize FOXO1-positive staining; a brighter image should be provided.

Response: Figure 1F has been replaced with a brighter image to allow clearer visualization of FOXO1-positive staining.

Minor Point 2: Figure S1 presents important findings and should be moved from the supplemental data to the main figures.

Response: We appreciate the reviewer’

---

## [Decision Letter · Decision Letter 1]

5 Jan 2026

Dear Dr. Daniel,

Thank you for submitting your manuscript to PLOS ONE. After careful consideration, we feel that it has merit but does not fully meet PLOS ONE’s publication criteria as it currently stands. Therefore, we invite you to submit a revised version of the manuscript that addresses the points raised during the review process.

**Thank you for submitting the following manuscript to PLOS ONE.**

**Please revise the manuscript according to the reviewers' comments and upload the revised file.**

We look forward to receiving your revised manuscript.

Kind regards,

Yung-Hsiang Chen, Ph.D.

Academic Editor

PLOS One

Journal Requirements:

Additional Editor Comments:

Thank you for submitting the following manuscript to PLOS ONE.

Please revise the manuscript according to the reviewers' comments and upload the revised file.

Reviewers' comments:

Reviewer's Responses to Questions

**Comments to the Author**

Reviewer #1: All comments have been addressed

Reviewer #2: (No Response)

2. Is the manuscript technically sound, and do the data support the conclusions?

Reviewer #1: Yes

Reviewer #2: Partly

3. Has the statistical analysis been performed appropriately and rigorously?

Reviewer #1: Yes

Reviewer #2: Yes

4. Have the authors made all data underlying the findings in their manuscript fully available?

Reviewer #1: Yes

Reviewer #2: Yes

5. Is the manuscript presented in an intelligible fashion and written in standard English?

Reviewer #1: Yes

Reviewer #2: Yes

Reviewer #1: (No Response)

Reviewer #2: While I appreciate the inclusion of TLR3 protein data, the manuscript needs to address more clearly how FOXO1 would modulate TLR3 signalling (or signalling via other viral PRRs/RLRs) if this modulation of transcription is not apparent at the protein level. If TLR3 expression was altered, the change in title would have been sufficient, however due to this null finding I still would recommend that additional experiments are completed to explore other ways in which FOXO1 could modulate cytokine release and viral signalling (ie. through regulation of other levels of PRR signalling p-TBK1, p-IRF3, NF-κB activation etc.), since the claims about TLR3 mRNA regulation, repressed cytokine release, as well as viral modulation are currently disconnected. Alternatively, claims about viral modulation via FOX01 could be isolated to a different manuscript, and this paper could focus solely on the regulation on TLR3 mRNA by FOXO1, however a more definitive EMSA or alternative way to show FOXO1 binding to DNA in airway epithelial cell lines would be required. As the manuscript currently stands I don’t believe “conclusions are presented in an appropriate fashion and are supported by the data”, as per the PLOS One publication standards, since the title implies that FOX01 coordinates antiviral responses through TLR3 dependent mechanism, and there is a lack of evidence for this.

.

Reviewer #1: **Yes:**Tomoh MatsumiyaTomoh MatsumiyaTomoh MatsumiyaTomoh Matsumiya

Reviewer #2: No

---

## [Author Response · Author response to Decision Letter 2]

17 Feb 2026

Rebuttal Letter – Full Revised Version (Round 2) - 20260217

Revised Title: “FOXO1 transcription factor modulates airway epithelial responses to viral infection”

Manuscript ID: PONE-D-25-41597R1 Journal: PLOS ONE

Dear Dr. Chen,

We thank you for the opportunity to submit a new revision of our manuscript entitled “FOXO1 regulates TLR3 mRNA expression and selective inflammatory responses to Poly(I:C) in airway epithelial cells” (Manuscript ID: PONE-D-25-41597R1).

Following the first revision, Reviewer #1 indicated that their concerns were fully addressed. Reviewer #2 raised some additional concerns regarding the validity of our conclusion that FOXO1 participates in TLR3 signaling in airway epithelial cells. We thank the reviewer for bringing up these issues and we believe we have addressed their concern in our revised manuscript. To this effect, we have added new immunofluorescence data demonstrating that Poly(I:C) stimulation induces FOXO1 nuclear localization in BEAS-2B cells, consistent with stimulus-dependent FOXO1 activation. To clarify the distinction between FOXO1’s effects on TLR3 expression and its effects on downstream TLR3-activated pathways, we reorganized the Results and Discussion sections to improve clarity and reduce potential confusion.

We have also further revised the title, abstract, and discussion to ensure that all conclusions remain strictly aligned with the experimental evidence and to avoid mechanistic overstatement. A detailed response is provided below.

Sincerely,

Nadia M. Daniel, MSc

(on behalf of all authors)

Reviewer #2 Comment: “While I appreciate the inclusion of TLR3 protein data, the manuscript needs to address more clearly how FOXO1 would modulate TLR3 signalling (or signalling via other viral PRRs/RLRs) if this modulation of transcription is not apparent at the protein level. If TLR3 expression was altered, the change in title would have been sufficient, however due to this null finding I still would recommend that additional experiments are completed to explore other ways in which FOXO1 could modulate cytokine release and viral signalling (ie. through regulation of other levels of PRR signalling p-TBK1, p-IRF3, NF-κB activation etc.), since the claims about TLR3 mRNA regulation, repressed cytokine release, as well as viral modulation are currently disconnected. Alternatively, claims about viral modulation via FOX01 could be isolated to a different manuscript, and this paper could focus solely on the regulation on TLR3 mRNA by FOXO1, however a more definitive EMSA or alternative way to show FOXO1 binding to DNA in airway epithelial cell lines would be required. As the manuscript currently stands I don’t believe “conclusions are presented in an appropriate fashion and are supported by the data”, as per the PLOS One publication standards, since the title implies that FOX01 coordinates antiviral responses through TLR3 dependent mechanism, and there is a lack of evidence for this.”

Response to Reviewer #2:

We understand the reservations of the reviewer regarding the role of FOXO1 in TLR3 signaling. To support the conclusion that FOXO1 participates in the epithelial response to TLR3 activation, we performed immunofluorescence experiments demonstrating increased FOXO1 nuclear localization following Poly(I:C) stimulation (Fig. 5A+B). We therefore propose that altered outcomes of TLR3 activation (i.e. increased TLR3 mRNA expression and increased release of certain cytokines) in the absence of FOXO1 are most likely the result of lack of TLR3-induced FOXO1 activation and probably not the result of altered TLR3 expression. We do not claim definitive FOXO1 control of canonical downstream signaling pathways or direct binding to the TLR3 promoter.

Unfortunately, further studies on the effects of FOXO1 downregulation on signaling pathways activated by TLR3 are difficult to execute at this time, as we do not yet know which specific pathways are affected or whether FOXO1 acts upstream or downstream of other proteins in these pathways. Identifying these pathways is a major component of our future plans for this project, but we feel that pursuing this level of pathway dissection would be beyond the scope of the current manuscript.

In addition, FOXO1 deficiency alters epithelial barrier properties, suggesting a mechanism by which FOXO1 may influence host–virus interactions at the epithelial surface. Disruption of epithelial barrier properties can affect viral access to underlying immune compartments and modulate local inflammatory responses. In line with this epithelial-focused role, our SARS-CoV-2 data indicate that FOXO1 participates in epithelial responses activated during viral infection. Together, these observations support a model in which FOXO1 selectively shapes epithelial inflammatory responses through effects on both transcriptional programs and barrier properties, rather than acting as a global regulator of antiviral signaling.

To clarify the distinction between FOXO1’s effects on TLR3 expression and its effects on downstream TLR3-activated pathways as well as other anti-viral pathways, we reorganized the Results and Discussion sections to improve clarity. We believe that our conclusions are now supported by the data.

Following the reviewer’s guidance and to better align the title with the data presented, we propose revising the manuscript title from “FOXO1 transcription factor regulates Toll-like receptor 3 mRNA and coordinates antiviral responses in BEAS-2B airway epithelial cells” to “FOXO1 transcription factor modulates airway epithelial responses to viral infection.”

---

## [Decision Letter · Decision Letter 2]

3 Mar 2026

FOXO1 transcription factor modulates airway epithelial responses to viral infection.

PONE-D-25-41597R2

Dear Dr. Daniel,

We’re pleased to inform you that your manuscript has been judged scientifically suitable for publication and will be formally accepted for publication once it meets all outstanding technical requirements.

Kind regards,

Yung-Hsiang Chen, Ph.D.

Academic Editor

PLOS One

Additional Editor Comments (optional):

Congratulations on the acceptance of your manuscript, and thank you for your interest in submitting your work to PLOS ONE.

Reviewers' comments:

Reviewer's Responses to Questions

**Comments to the Author**

Reviewer #3: All comments have been addressed

2. Is the manuscript technically sound, and do the data support the conclusions?

Reviewer #3: Yes

3. Has the statistical analysis been performed appropriately and rigorously?

Reviewer #3: Yes

4. Have the authors made all data underlying the findings in their manuscript fully available?

Reviewer #3: Yes

5. Is the manuscript presented in an intelligible fashion and written in standard English?

Reviewer #3: Yes

Reviewer #3: This paper investigates the role of the transcription factor FOXO1 in airway epithelial responses to viral infection, showing that FOXO1 regulates Toll-like receptor 3 (TLR3) transcription, selectively modulates cytokine release (notably IL-6 and CCL2), and influences epithelial barrier repair. Using BEAS-2B and primary human bronchial epithelial cells, the authors demonstrate that FOXO1 knockdown or inhibition reduces TLR3 mRNA levels, alters wound healing dynamics, and decreases SARS-CoV-2 replication, while FOXO1 overexpression enhances TLR3 transcription. The findings suggest FOXO1 acts as a selective regulator of antiviral defense and inflammation in airway epithelium, highlighting its potential as a therapeutic target, though the precise mechanisms of promoter binding and downstream signaling remain unresolved.

The manuscript has been thoroughly revised in response to the reviewers’ comments, and the current version reflects significant improvements in clarity, methodology, and presentation. Both reviewers’ concerns have been carefully addressed, and the revisions strengthen the scientific rigor and readability of the paper. Overall, the study is now well-prepared and suitable for publication, and I recommend acceptance in its present form.

.

Reviewer #3: No

---

## [Editor Report · Acceptance letter]

PONE-D-25-41597R2

PLOS One

Dear Dr. Daniel,

I'm pleased to inform you that your manuscript has been deemed suitable for publication in PLOS One. Congratulations! Your manuscript is now being handed over to our production team.

Kind regards,

on behalf of

Dr. Yung-Hsiang Chen

Academic Editor

PLOS One